# Untargeted Metabolomics Analysis Revealed the Major Metabolites in the Seeds of four *Polygonatum* Species

**DOI:** 10.3390/molecules27041445

**Published:** 2022-02-21

**Authors:** Jianjun Qi, Jianhe Wei, Dengqun Liao, Zimian Ding, Xia Yao, Peng Sun, Xianen Li

**Affiliations:** Key Laboratory of Bioactive Substances and Resources Utilization of Chinese Herbal Medicine, Ministry of Education, Institute of Medicinal Plants Development, Chinese Academy of Medical Sciences, Peking Union Medical College, Beijing 100193, China; wjianh@263.net (J.W.); dqliao@implad.ac.cn (D.L.); zmding@implad.ac.cn (Z.D.); yaoxia1118@163.com (X.Y.); psun@implad.ac.cn (P.S.); xeli@implad.ac.cn (X.L.)

**Keywords:** *Polygonatum*, seed, metabolite, untargeted metabolome, UPLC-QTOF-MS/MS

## Abstract

Most *Polygonatum* species are widely used in China as a source of medicine and food. In this study, a UPLC-QTOF-MS/MS system was used to conduct an untargeted metabolomics analysis and compare the classes and relative contents of metabolites in the seeds of four *Polygonatum* species: *P. sibiricum* (Ps), *P. cyrtonema* (Pc), *P. kingianum* (Pk), and *P. macropodium* (Pm). The objectives of this study were to clarify the metabolic profiles of these seeds and to verify their medicinal and nutritional value via comparative analyses. A total of 873 metabolites were identified, including 185 flavonoids, 127 lipids, 105 phenolic acids, and 36 steroids. The comparative analysis of metabolites among *Polygonatum* seed samples indicated that flavonoids, steroids, and terpenoids were the main differentially abundant compounds. The results of principal component analysis and hierarchical clustering were consistent indicating that the metabolites in Ps and Pm are similar, but differ greatly from Pc and Pk. The data generated in this study provide additional evidence of the utility of *Polygonatum* seeds for producing food and medicine.

## 1. Introduction

Polygonati rhizoma (HuangJing in Chinese) was originally used as a Taoist health food rather than as medicine in ancient China, but Ming Yi Bie Lu, which is a book listing herbs, described its medicinal value about 2000 years ago [1]. Polygonati rhizoma, which is derived from the rhizome of three plants from the genus *Polygonatum* (i.e., *P. sibiricum*, *P. cyrtonema*, and *P. kingianum*), has many pharmacological effects for tonifying *Qi* and *Yin* as well as for treating age-related diseases, diabetes, lung diseases, fatigue, weakness, and indigestion [2,3,4,5,6,7]. Most *Polygonatum* plant species have properties that make them useful as medicine and food in China. More specifically, 38 *Polygonatum* species have been used to produce traditional medicines and functional foods in China [6]. A previous investigation of the chemical constituents of aerial plant parts (i.e., stem, leaves, and fruits) revealed *Polygonatum verticillatum* contains active components with strong antibacterial effects [8]. During the Song Dynasty more than 1000 years ago, the edibility of *Polygonatum* plants (i.e., roots, leaves, flowers, and fruits) was confirmed, with the greatest medicinal effects detected for flowers, followed by fruits and rhizomes. Huang et al. [9] recently identified the following five key compounds in *P. sibiricum* flowers on the basis of a UPLC-QTOF-MS analysis: wogonin, rhamnetin, dauriporphine, chrysosplenetin B, and 5-hydroxy-7,8-panicolin, and revealed its antitumor activity. Flowers and rhizomes of *P. cyrtonema* and *P. filipes* both had polysaccharides, saponins, amino acids, total phenols, mineral elements, but the amino acids content and essential trace element content in flowers is higher than that in rhizomes [10]. Shoots of *P. cyrtonema* had polysaccharides, proteins, amino acids, and total phenols, and its essential amino acids account for 35.57–39.44% of the total amino acids content, which was close to the standard of the ideal protein proposed by FAO/WHO (the essential amino acid /total amino acid is about 40%) [11]. Si and Zhu [12] reported that Polygonati rhizoma is rich in polysaccharide, fructose-oligosaccharides and fructose (about 50%), protein (11.16%), 7 kinds of essential amino acids, 110 kinds of steroidal saponins (3.16–5.05%), and flavonoids (0.66–1.91%), and can be used as a new high-quality crop with great potential in understory crop production and healthy food development. At present, there are no reports on the chemical constituents and edible properties of the *Polygonatum* seeds. Metabolomics is a rapidly developing technology that has effectively contributed to many plant-related sciences and the identification of bioactive substances with medicinal uses [13,14,15]. Untargeted metabolomics profiles are useful for qualitative and quantitative analyses because many groups of metabolites can be examined and specific compounds can be characterized. Accordingly, in this study, a UPLC-QTOF-MS/MS method was used to explore the metabolites and the differentially abundant compounds among the seeds of four *Polygonatum* species.

The four examined *Polygonatum* species (*P. sibiricum*, *P. cyrtonema*, *P. kingianum*, and *P. macropodium*) were selected because they grow in diverse ecological regions in China (Figure 1). Two of the species produce leaves in whorls (*P. sibiricum* and *P. kingianum*), which is in contrast to the alternate leaves of *P. cyrtonema* and *P. macropodium* plants. Additionally, the perianths are pink in *P. kingianum*, yellowish green in *P. cyrtonema*, and milky white in *P. sibiricum* and *P. macropodium*. Despite these differences, *P. sibiricum*, *P. cyrtonema*, and *P. kingianum* are included in the Pharmacopeia of China as the original medicinal herbs with a rhizome. In this study, *P. macropodium*, which has traditionally been used as a medicinal herb and for cooking, was analyzed because it grows in the same habitat as *P. sibiricum*, but its seeds differ in terms of size and appearance. For our metabolomics analysis, we used a UPLC-QTOF-MS/MS system to explore the seed chemical compositions and to verify the medicinal value or edibility.

## 2. Results

### 2.1. Metabolite Profiles

In this study, an analysis of the Ps1 (*P. sibiricum*), Ps2 (*P. sibiricum*), Pc (*P. cyrtonema*), Pk (*P. kingianum*), and Pm (*P. macropodium*) seed samples using a UPLC-ESI-MS/MS system detected 873 metabolites (data have been de-duplicated), which were divided into the following classes: flavonoids, lipids, phenolic acids, amino acids and derivatives, organic acids, alkaloids, nucleotides and derivatives, steroids, terpenoids, lignans and coumarins, tannins, quinones, and others (Table 1). Among the most common compounds were 86 flavonoids, 32 flavonols, 18 isoflavones, 18 flavonoid carbonosides, 12 dihydroflavone flavanols, 6 chalcones, 3 dihydroflavonols, and 1 sinensetin (Table 2). The specific steroidal saponins identified in the *Polygonatum* seeds included polygonatumoside A, polygonatone C, kingianoside B, neosibiricoside D, polygonatoside D, kingianoside G, kingianoside I, and polygonatumoside F. In contrast to the considerable differences in the flavonoid, steroid, and terpenoid contents among the five seed samples, there were relatively few differences in the number and type of metabolites in the following six classes: amino acids and derivatives, nucleotides and derivatives, lignans and coumarins, tannins, lipids, and others (Table 1). Eight steroidal saponins were common to all seed samples, namely, polygonatumoside A, polygonatone C, pennogenin-3-*O*-glucoside, diosgenin, polygonatoside D, polygonatumoside F, aculeatiside A, and (3β,14α)-3-*O*-β-d-glucopyranosyl-(1→2)-[β-d-xylopyranosyl-(1→3)]-β-d-glucopyranosyl-(1→4)-β-d-galactopyranoside-yamogenin (Table 2). Additionally, the 31 detected terpenoids comprised 20 triterpene saponins, 6 sesquiterpenoids, 3 diterpenoids, and 2 monoterpenoids. Specific details regarding the identified compounds are provided in Table 1 and Table 2.

### 2.2. Metabolite Analyses via PCA as well as Hierarchical Cluster and Correlation Analyses

A PCA (Principal component analysis) involving several principal components may be useful for determining the overall metabolic differences among the sample groups and the variability between the intra-group samples. In this study, the first principal component (PC1) accounted for 36.41% of the variation, whereas the second principal component (PC2) accounted for 21.52% of the variation. Additionally, Ps1, Ps2, and Pm clustered together, separate from Pc and Pk (Figure 2a), implying the five seed samples were divided into the following three groups according to their metabolites: (1) Ps1, Ps2, and Pm, (2) Pc, and (3) Pk. However, in the three-dimensional PCA plot, the third principal component (PC3), which accounted for 12.19% of the variation, indicated that Pm did not cluster with Ps1 and Ps2, indicative of four groups for the five seed samples (Figure 2b). Furthermore, the PCA results revealed the relatively low variability among biological replicates, which reflected the strong correlation between replicates.

The metabolite content data were normalized by unit variance scaling. A heatmap was drawn using the pheatmap package of the R software. Moreover, the metabolite accumulation patterns of the different samples underwent a cluster analysis (Figure 3). The cluster lines on the left side of Figure 3 represent metabolite clusters. There were three metabolite clusters from top to bottom (i.e., C1, C2, and C3). Cluster C1 included three subclusters, namely, C11 (high Pm metabolite content) and C12 and C13 (high Ps1 and Ps2 metabolite contents). Obvious features of C2 and C3 were, respectively, the relatively high Pc and Pk metabolite contents. The top cluster of Figure 3 revealed four levels. The Ps1 and Ps2 samples had the closest relationship, with the next closest relationships involving Pm, Pc, and then Pk. This clustering is consistent with the ecogeographic distribution of these species.

The reproducibility between samples within a group was assessed by a correlation analysis. A high correlation coefficient indicated the reproducibility between intra-group samples (i.e., reliability of the analysis). The results of the correlation analysis confirmed the high reproducibility between samples. Furthermore, the Ps1, Ps2, and Pm samples were highly correlated (Figure 1).

### 2.3. OPLS-DA Results

The OPLS-DA is an effective method for screening differentially abundant metabolites because it can maximize the differences between groups [16]. The prediction parameters for the OPLS-DA model included R2X, R2Y, and Q2. Specifically, R2X and R2Y respectively represent the interpretation rate for the X and Y matrices, whereas Q2 reflects the utility of the model for making predictions. For all three parameters, a value close to one indicates a stable and reliable model. Additionally, Q2 > 0.5 indicates an effective model, but Q2 > 0.9 suggests the model is ideal for analyses. In this study, the OPLS-DA model was used for eight pairwise comparisons of metabolite contents. The R2X value ranged from 0.650 to 0.834, the R2Y value was close to 1, and the Q2 value ranged from 0.868 to 0.990. Thus, the model was stable and reliable, making it useful for screening and analyzing differentially abundant metabolites (Table 3).

### 2.4. Analysis of Differentially Abundant Metabolites among the Seed Samples

On the basis of the OPLS-DA (OSC partial least squares–discriminant analysis,), the variable importance in projection (VIP) value of the multivariate analysis model can be used to initially screen for the differentially abundant metabolites among the species or groups. Moreover, the p-value and fold-change in the univariate analysis were combined to further screen for differentially abundant metabolites. The criteria used for detecting differentially abundant metabolites were as follows: fold-change ≥ 2 and ≤ 0.5 and VIP ≥ 1. In this study, eight pairwise comparisons of seed sample metabolite contents according to the fold-change and VIP value were performed. The results revealed significant differences between Ps (Ps1 and Ps2) and Pc and Pk, with differences reaching 51.76–61.34% (Table 4). The smallest difference was detected in the Ps1 vs. Ps2 comparison (27.22%; of the 812 compounds, 65 and 156 had decreased and increased abundances, respectively).

After the qualitative and quantitative analyses, the detected metabolites were grouped and compared. Table 3 lists the top 10 differentially abundant metabolites according to the log2(fold-change) for each comparison. Steroidal saponins, flavonoids, and terpenoids were the three bioactive compounds with the greatest differences among the comparisons. In the Ps1 vs. Pc and Ps2 vs. Pc comparisons, the top 25 compounds included 15 steroidal saponins and 5 flavonoids. Additionally, eight and seven identical compounds in these two comparisons were among the top 10 compounds with decreased and increased abundances, respectively (bold numbers in column four of Table 3). In the Ps1 vs. Pc and Ps2 vs. Pc comparisons, the top 27 compounds included seven steroidal saponins, seven flavonoids, and four terpenoids. The Pk vs. Pc comparison revealed 13 steroidal saponins and 3 triterpenes among the top 20 compounds.

The k-means cluster analysis indicated that 781 metabolites were clustered in nine subclasses according to the standardized relative metabolite contents (Figure 4 and Table 4). Subclasses 1, 2, 6, and 9 comprised 536 metabolites (68.63%). The contents of the metabolites in these subclasses varied in the five examined seed samples. For example, the subclass 1 metabolite contents in Pk differed from the corresponding contents in Ps1, Ps2, Pc, and Pm. The subclass 2 and 6 metabolite contents in Pc and Pk differed from the corresponding abundances in Ps1, Ps2, and Pm. Furthermore, 150 metabolites (53 flavonoids) in subclass 9 were exclusive to Pc.

The differentially abundant metabolites detected in each comparison were annotated according to the Kyoto Encyclopedia of Genes and Genomes (KEGG) database (Figure 5). The KEGG classification results and enrichment analysis indicated that the differentially abundant metabolites revealed by eight comparisons were associated with metabolism, environmental information processing, and genetic information processing, including 53–84 metabolic pathways (mostly belonging to the metabolism category) (Figure 5). The five most enriched KEGG pathways were metabolic pathways, biosynthesis of secondary metabolites, biosynthesis of amino acids, 2-oxocarboxylic acid metabolism, and flavonoid, isoflavonoid, flavone, and flavonol biosynthesis (Figure 5).

## 3. Discussion

The *Polygonatum* rhizome has traditionally been used as a source of medicine, unlike the rest of the plant. More than 1000 years ago, Tang Shenwei recorded the medicinal value of *P. sibiricum* rhizomes, leaves, flowers, and fruits in his book Zheng Lei Ben Cao (1097–1108 CE during the Song Dynasty) [1]. He demonstrated the medicinal effects were greatest for the flowers, followed by the fruits and then the rhizomes. Zhao et al. [6] reviewed studies on the chemical constituents of *Polygonatum* plants and confirmed that 37 *Polygonatum* species, including one variety, have been used as a traditional medicine or a functional food. Among these species, all *P. verticillatum* and *P. nodosum* plant parts, including the seeds, are harvested for medicinal use, whereas for the other species, only the rhizome is considered to have useful medicinal properties. A recent study regarding the chemical composition of *P. sibiricum* flowers identified 64 components (e.g., flavonoids, alkaloids, terpenoids, saponins, and organic acids), of which 35 had potential antitumor activities [11]. Investigations on the nutrient contents of *P. cyrtonema* flowers and shoots proved that the flower and shoot polysaccharide contents were 50% or 33% of that in the rhizomes, but the saponin, total phenol, and amino acid contents were similar or several times higher in the flowers and shoots [10,11]. In the current study, we identified 873 metabolites in the seeds of four *Polygonatum* species, including many bioactive components (e.g., 185 flavonoids, 105 phenolic acids, 88 amino acids and derivatives, 74 organic acids, 54 alkaloids, 36 steroids, 31 terpenoids, and 24 lignans and coumarins) (Table 1 and Table 2). The main active components of Polygonati Rhizoma are reportedly steroidal saponins, triterpenoid saponins, flavonoids, polysaccharides, and lectins [6,17,18,19,20]. Our research demonstrated that there are many bioactive components in *Polygonatum* seeds, which may be relevant for clarifying the medicinal and nutritional value of the seeds and further analysis of the specific bioactive components in the seeds is needed in the future.

Plant seeds are reproductive organs, but they are also the source of human food or medicine. Normally, a seed consists of an embryo and endosperm and its main chemical components are starch, proteins, fat, sugars, and minerals. *Polygonatum* seeds contain an underdeveloped embryo occupying only about 1% of the seed volume, with the endosperm filling most of the remaining volume [21]. Therefore, most of the seed bioactive substances may be contained in the endosperm. The seed setting rate was highest for *P. sibiricum*, with each plant able to produce about 15 g seeds (dry weight); however, our preliminary analysis indicated that the seed yield may be 1–2-fold higher for *P. kingianum* than for *P. sibiricum* (data not shown). These findings may be relevant for increasing the seed yield in future studies, which is an important research goal.

Metabolome analyses cannot determine the absolute content of each metabolite, but a comparative analysis of samples on the basis of relative metabolite contents is useful for identifying biologically active metabolites [22,23]. Moreover, a VIP > 1.0 and a fold-change > 2 or < 0.5 may be set as the thresholds for detecting differentially abundant metabolites. In the present study, eight pairwise comparisons were conducted using these thresholds to analyze the metabolite contents in the seeds collected from four *Polygonatum* species. Significant differences were revealed, especially in the Ps vs. Pc, Ps vs. Pk, and Pk vs. Pc comparisons (Table 4). The log2(fold-change) values indicated the significantly differentially abundant metabolites among the sample groups were primarily steroidal saponins, flavonoids, and terpenoids, which contributed to the diversity in the nutritional or medicinal value of the seeds (Table 3).

The *Polygonatum* plants are distributed in different climatic zones and have high genetic diversity. *P. sibiricum*, *P. cyrtonema*, and *P. kingianum* are mainly distributed in the northern warm temperate, subtropical, and subtropical Yunnan–Guizhou plateau regions, respectively, whereas *P. macropodium* grows in the northern region of the northern temperate zone (Figure 1). The phylogenetic analysis showed that there are three sections within the tribe Polygonateae, Sibirica, Polygonatum, and Verticillata [24,25]. This classification suggests *P. sibiricum*, *P. cyrtonema*, and *P. kingianum* might belong to the Sibirica, *Polygonatum*, and Verticillata sections, respectively. In this study, the comparative study on metabolites among Polygonatum seeds indicated that several seeds belonging to the same genus may have different bioactive components especially in flavonoids, steroids, and terpenoids. These differences require us to analyze the specific component of each seed and provide more useful medicinal or edible value in the future.

## 4. Materials and Methods

### 4.1. Plant Materials and Sample Preparation

Five seed samples were collected from four *Polygonatum* species in the fall of 2019 from the following four locations in China: Ps1, Xiuwu county, Henan province (113°20′ E, 35°27′ D, H 720 m); Ps2 and Pm, the Institute of Medicinal Plant Development, Beijing (116°16′ E, 40°2′ D, H 60 m); Pc, Chun’an county, Zhejiang province (119°10′ E, 29°58′ D, H 235 m); and Pk, Yimen county, Yunnan province (102°5′ E, 24°41′ D, H 2,180 m). The seed samples were dried under natural conditions in a shade room and stored in a nylon mesh bag at room temperature.

The seed samples were crushed for 1.5 min using the MM400 mixer mill (Retsch, Laichi, Germany) and zirconia beads. The seed powder (100 mg) was resuspended in 1.2 mL 70% methanol and rotated six times (30 s each time) with 30 min intervals. The samples were incubated overnight at 4 °C. They were then centrifuged at 12,000 rpm for 10 min, after which the extracts were filtered through the SCAA-104 membrane (0.22 μm pores; ANPEL, Shanghai, China, http://www.anpel.com.cn/ accessed on 17 February 2022) before the UPLC-MS/MS analysis.

### 4.2. UPLC-ESI-Q TRAP-MS/MS Analysis

The sample extracts were analyzed using a UPLC-ESI-MS/MS system (UPLC, Nexera X2, Shimadzu, Kyoto, Japan; MS, 4500 Q TRAP, Applied Biosystems, Foster City, CA, USA). The analytical conditions were as follows: UPLC column, Agilent SB-C18 (1.8 µm, 2.1 mm × 100 mm); mobile phase, solvent A (pure water with 0.1% formic acid) and solvent B (acetonitrile with 0.1% formic acid). Sample analyses were performed using the following gradient: 95% A, 5 % B; 5% A, 95% B (linear gradient) within 9 min and then maintained for 1 min; and 95% A, 5.0% B within 1.10 min and then maintained for 2.9 min. The flow rate was 0.35 mL/min. The column temperature was set to 40 °C and the injection volume was 4 μL. The effluent was alternatively connected to an ESI-triple quadrupole-linear ion trap (Q TRAP) mass spectrometer. The linear ion trap and triple quadrupole scans were completed using the AB4500 Q TRAP mass spectrometer in the positive and negative ion modes and the Analyst 1.6.3 software (AB Sciex). The ESI source operation parameters were set as follows: ion source, turbo spray; source temperature, 550 °C; ion spray voltage, 5500 V (positive ion mode)/−4500 V (negative ion mode); ion source gas I, gas II, and curtain gas, 50, 60, and 25.0 psi, respectively; and collision-induced dissociation, high. The instrument tuning and mass calibration were performed using 10 and 100 μmol/L polypropylene glycol solutions in triple quadrupole and linear ion trap modes, respectively. The triple quadrupole scans were acquired during MRM experiments with the collision gas (nitrogen) set to medium. The DP and CE were optimized for individual MRM transitions. For each period, a specific set of MRM transitions was monitored according to the eluted metabolites. The quality control sample was injected periodically throughout the analytical run to monitor the UPLC-MS/MS system.

### 4.3. Qualitative and Quantitative Analyses of Metabolites

The metabolites were qualitatively and quantitatively analyzed as previously described [26]. On the basis of the MWDB database (Metware Biotechnology Co., Ltd., Wuhan, China) and the public database comprising metabolite information, the sample metabolites were qualitatively and quantitatively analyzed by mass spectrometry. The characteristic ions of each substance were screened using the triple quadrupole rod, and the signal strengths of the characteristic ions were determined using the detector. The mass spectrometry file for each sample was analyzed using the MultiQuant software (version 3.0.3) to integrate and correct chromatographic peaks as well as to determine the relative content of the corresponding substances according to the peak area. Finally, all chromatographic peak area data were obtained. To compare the metabolite contents among samples, we calibrated the mass spectrum peaks for each metabolite in different samples on the basis of the metabolite retention time and peak pattern. Thus, the accuracy of the qualitative and quantitative analyses was ensured.

### 4.4. Statistical Analysis

Each analysis was completed using three biological replicates. The cluster analysis, principal component analysis (PCA), and orthogonal signal correction and partial least squares–discriminant analysis (OPLS-DA) were performed using R (http://www.r-project.org/ accessed on 17 February 2022) as previously described [27].

## 5. Conclusions

In this study, we first investigated the metabolites of *Polygonatum* seeds and elucidated the chemical compositions and differences among species. A total of 873 metabolites that were divided into 13 classes were included in the metabolite profiles of 5 seed samples following a UPLC-ESI-MS/MS analysis. A comparison of the metabolomes revealed distinct features among the examined seed samples. There were almost no differences in the contents of the metabolites in the amino acids and derivatives, nucleotides and derivatives, and others (e.g., saccharides, alcohols, and vitamins) classes among the seed samples. In contrast, there was some diversity in the contents of lipids, phenolic acids, lignans and coumarins, tannins, and quinones among the seed samples. The flavonoid, steroid, and terpenoid classes and contents varied among the *Polygonatum* seed samples; these compounds have relatively strong pharmacological effects. These findings indicate that different *Polygonatum* seeds differ in terms of their medicinal and nutritional value. Furthermore, rapid increases in the *Polygonatum* cultivation area will likely lead to substantial increases in the seed yield in the near future. Hence, our result is helpful for optimizing the use of the harvested seeds as medicinal and functional foods.

## Figures and Tables

**Figure 1 molecules-27-01445-f001:**
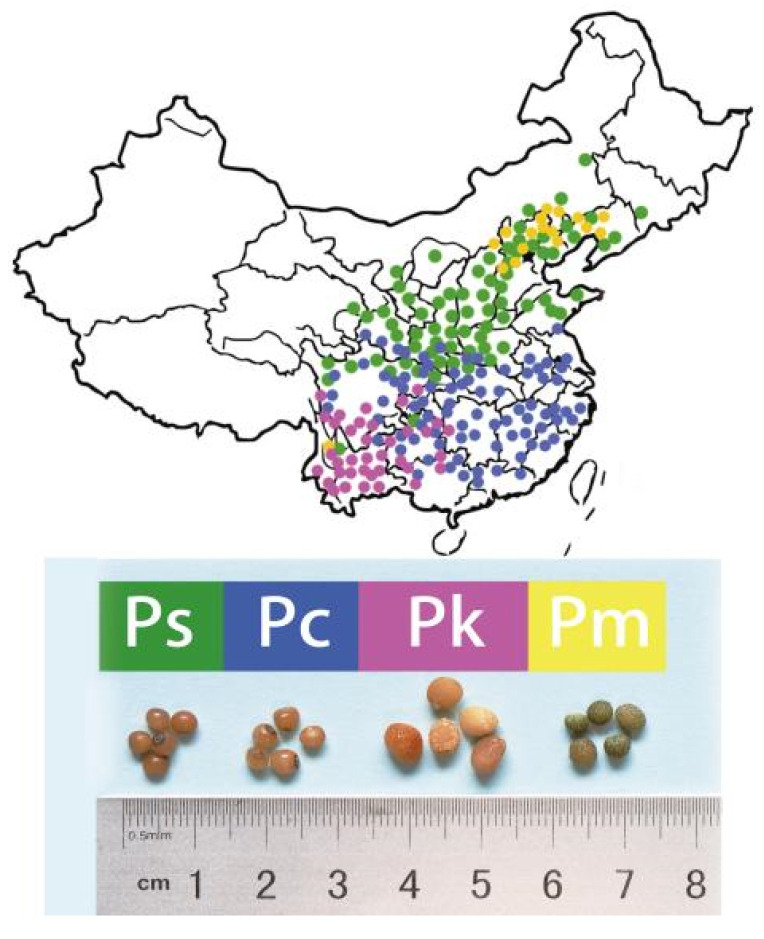
Distribution of four *Polygonatum* species in China (up) and the appearance of the seeds from the examined plants (down). Ps, *Polygonatum sibiricum*; Pc, *P. cyrtonema*; Pk, *P. kingianum*; Pm, *P. macropodium*.

**Figure 2 molecules-27-01445-f002:**
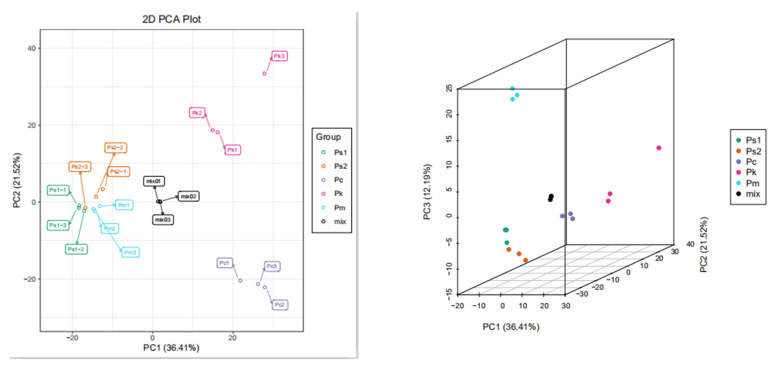
Differentially abundant metabolite analysis on the basis of a principal component analysis (PCA): Left: PCA score plot. Right: Three-dimensional PCA plot.

**Figure 3 molecules-27-01445-f003:**
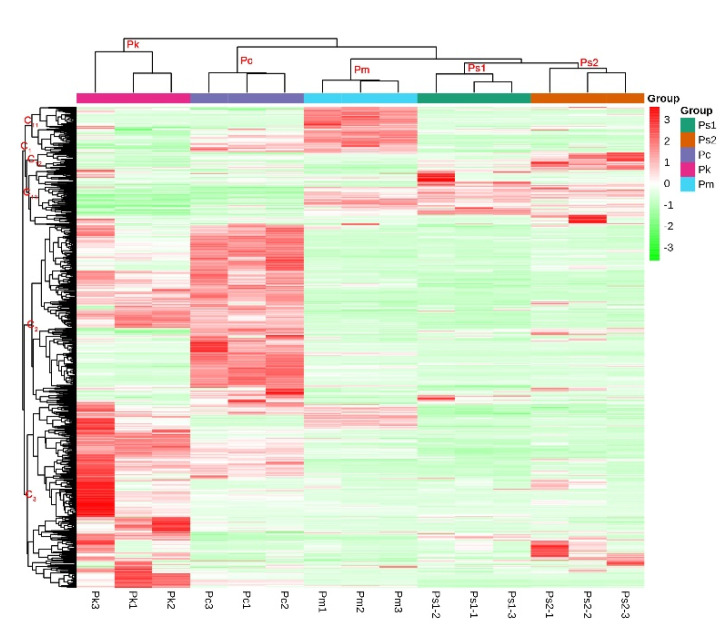
Cluster heatmap of the seed samples and metabolite classification. The cluster lines on the left represent the metabolite clusters. The cluster lines at the top represent sample clusters. Red and green indicate high and low metabolite contents, respectively.

**Figure 4 molecules-27-01445-f004:**
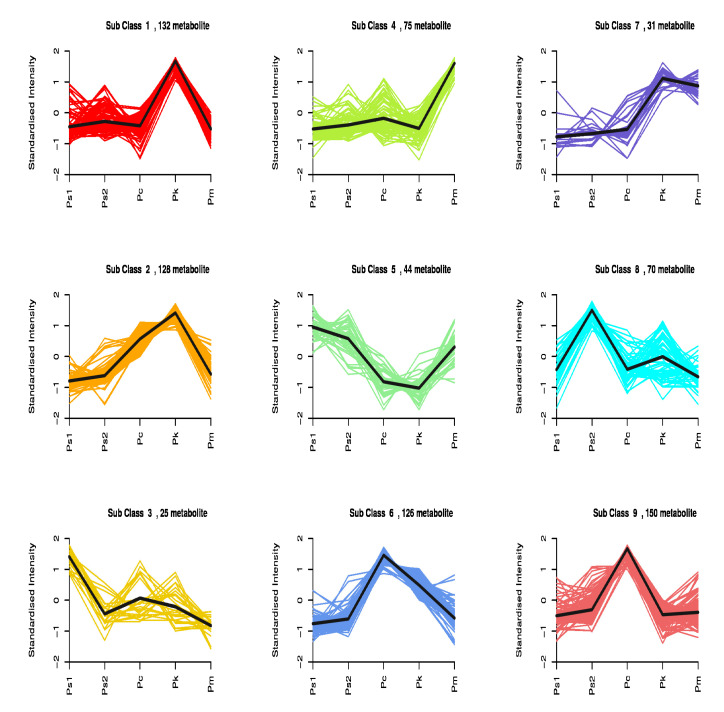
k-means cluster analysis of differential metabolites.

**Figure 5 molecules-27-01445-f005:**
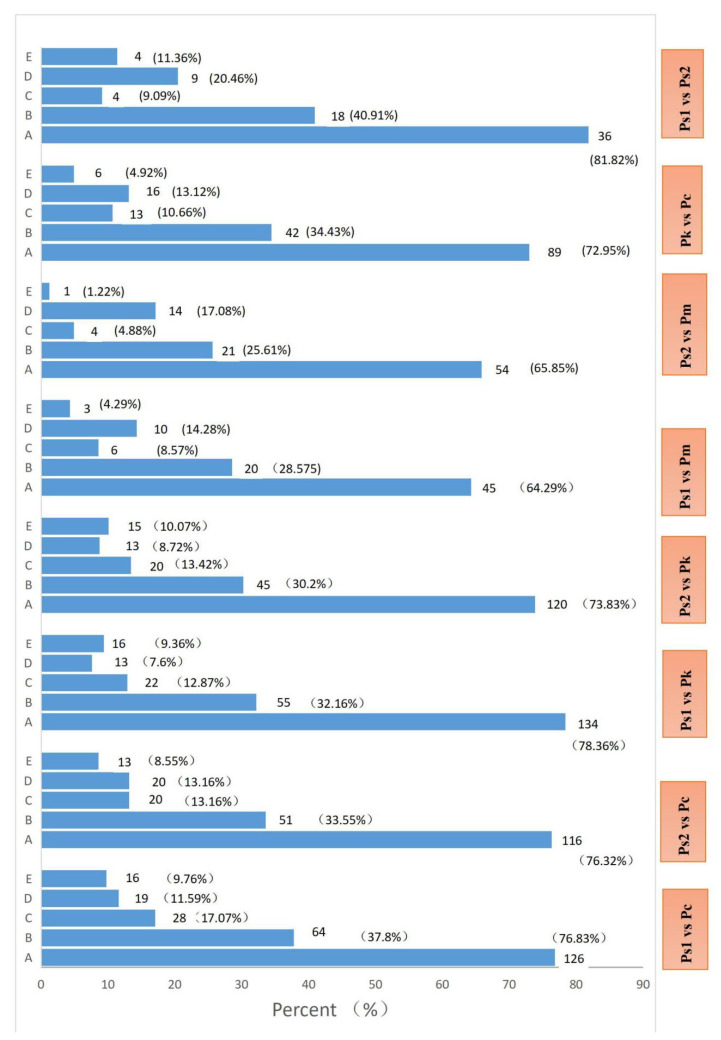
Five enriched KEGG pathways among the differentially abundant metabolites detected in eight comparisons (**A,** Metabolic pathways; **B**, Biosynthesis of secondary metabolites; **C**, Biosynthesis of amino acids; **D**, Flavonoid, isoflavonoid, flavone, and flavonol biosynthesis; **E**, 2-Oxocarboxylic acid metabolism). The abscissa presents the proportion of metabolites assigned to a pathway.

**Table 1 molecules-27-01445-t001:** Classification of the metabolites detected in five seed samples.

	Ps1	Ps2	Pc	Pk	Pm	Total Classes
Flavonoids	160	160	167	159	149	185
Lipids	116	117	122	124	123	127
Phenolic acids	93	98	100	101	97	105
Amino acids and derivatives	82	86	88	88	85	88
Organic acids	66	66	72	71	69	74
Alkaloids	50	56	54	53	54	57
Nucleotides and derivatives	45	46	46	46	46	46
Steroids	26	19	25	24	21	36
Terpenoids	22	22	21	27	22	31
Lignans and coumarins	22	21	22	20	21	24
Tannins	7	7	8	5	8	8
Quinones	1	1	3	3	3	4
Others	85	85	88	84	86	88
Total metabolites	775	782	816	805	784	873

**Table 2 molecules-27-01445-t002:** Identification of steroidal saponins in the seeds of four *Polygonatum* species and pairwise comparisons of the metabolites.

Index	Q1 (Da)	Q3 (Da)	Molecular Weight (Da)	Compounds	Ps1	Ps2	Pc	Pk	Pm	Ps1 vs. Pc	Ps2 vs. Pc	Ps1 vs. Pk	Ps2 vs. Pk	Ps1 vs. Pm	Ps2 vs. Pm	Pk vs. Pc	Ps1 vs. Ps2
Jmyn6479	3.29E + 02	1.39E + 02	3.30E + 02	Polygonatone C	+	+	+	+	+	insig	8.27	insig	9.08	insig	4.81	insig	−5.22
Zmcp010827	4.15E + 02	2.71E + 02	4.14E + 02	Diosgenin	+	+	+	+	+	insig	insig	5.6	8.69	−3.03	insig	−6.63	−3.09
mws1620	4.31E + 02	2.87E + 02	4.30E + 02	Ruscogenin	+	+	+	−	+	8.05	9.58	−11.09	−9.56	4.55	6.08	19.14	−1.53
pmp001034	4.33E + 02	2.89E + 02	4.32E + 02	Markogenin	−	−	+	−	−	12.17	12.17					12.17	
Zmcp008957	4.47E + 02	4.29E + 02	4.46E + 02	24-Hydroxypennogenin	−	−	+	−	−	14.86	14.86					14.86	
Jmyp5224	5.93E + 02	4.13E + 02	5.92E + 02	3β,26-diol-25(R)-Δ5,20(22)-diene-furostan-26-*O*-β-d-glucopyranoside	−	−	+	+	−	11.39	11.39	11.5	11.5			insig	
Zmcp006626	5.93E + 02	4.13E + 02	5.92E + 02	Pennogenin-3-*O*-glucoside	+	+	+	+	+	−3.18	insig	4.68	5.08	insig	insig	−7.86	insig
pmp000750	7.09E + 02	5.65E + 02	7.08E + 02	Ruscogenin-1-*O*-xylosyl(1,2)fucoside	−	−	+	−	−	15.90	15.9	−9.92				15.9	
pmp000748	7.09E + 02	5.65E + 02	7.08E + 02	Ruscogenin-1-*O*-xylosyl(1,3)fucoside	−	−	+	+	−	19.45	19.45	8.69	8.69			10.76	
pmp000753	7.23E + 02	5.79E + 02	7.22E + 02	Diosgenin-3-*O*-rhamnosyl(1,2)glcoside	+	−	+	−	−	6.71	17.1	−10.4		−10.4		17.1	−10.4
pmp000752	7.23E + 02	5.79E + 02	7.22E + 02	Ruscogenin-1-*O*-rhamnosyl(1,2)fucoside	+	−	+	−	−	6.73	16.65	−9.92		−9.92		16.65	−9.92
Cmdp004421	7.37E + 02	5.75E + 02	7.36E + 02	kingianoside B	+	+	+	−	−	insig	4.37	−17.24	−12.51	−17.24	−12.51	16.88	−4.74
Jmyp4233	7.39E + 02	5.77E + 02	7.38E + 02	(25S)-spirost-5-en-3β-ol-3-*O*-β-d-glucopyranosyl-(1→4)-β-d-galactopyranoside	+	+	−	+	+	−19.11	−17.63	insig	1.89	−1.29	insig	−19.52	−1.48
pmp000754	7.39E + 02	5.95E + 02	7.38E + 02	HydroxyDiosgenin-hamnosyl(1,2)glucoside	−	−		+	+	20.26	20.26	9.43	9.43	10.55	10.55	10.83	
pmp000755	7.39E + 02	5.95E + 02	7.38E + 02	HydroxyYamogenin-rhamnosyl(1,2)glucoside	−	−	+	+	+	20.35	20.32	9.63	9.63	10.74	10.74	10.7	
Hmdp003064	7.69E + 02	6.07E + 02	7.68E + 02	Kingianoside H	−	−	−	+	−			14.52	14.52			−14.52	
pmp000756	7.81E + 02	6.19E + 02	7.80E + 02	Ruscogenin-1-*O*-carboxyglucosyl(1,2)rhamnoside	+	+	+	−	−	−19.11	−17.63	insig	1.89	−1.29	insig	−19.52	−1.48
Hmdp003524	8.85E + 02	7.23E + 02	8.84E + 02	Gracillin	+	+	−	+	+	20.26	20.26	9.43	9.43	10.55	10.55	10.83	
Zmcp005230	8.85E + 02	4.14E + 02	8.84E + 02	Pennogenin-3-*O*-rhamnosyl(1→4)rhamnosyl(1→2)glucoside	+	−	+	−	−	20.35	20.32	9.63	9.63	10.74	10.74	10.7	
Zmcp005184	8.87E + 02	7.07E + 02	8.86E + 02	24-Hydroxypennogenin-3-*O*-rhamnosyl(1→2)xyloyl(1→4)glucoside	+	+	−	+	+			14.52	14.52			−14.52	
Hmdp003423	9.01E + 02	7.39E + 02	9.00E + 02	Neosibiricoside D	−	−	+	+	−	16.25	16.25	11.21	11.21			5.04	
pmp000762	9.01E + 02	7.39E + 02	9.00E + 02	Ruscogenin-1-*O*-rhamnosyl(1,2)glucosyl(1,2)glucoside	−	+	+	−	+	20.40	8.35		−2.04	17.7	5.65	20.4	12.04
Hmdp003403	9.01E + 02	7.39E + 02	9.00E + 02	Polygonatoside D	+	+	+	+	+	2.06	4.65	insig	insig	−2.33	insig	1.22	−2.59
pmp001041	9.03E + 02	7.41E + 02	9.02E + 02	Timosaponin B-III	+	−	−			−8.75		4.56	13.31	−3.17	insig	−13.31	−8.75
Hmdp002882	9.15E + 02	4.29E + 02	9.14E + 02	(25S)-Pratioside D1	+	−	+	−	−	3.62	15.96	−12.34		−12.34		15.96	−12.34
pmp001042	9.19E + 02	7.57E + 02	9.18E + 02	Timosaponin D	+	+	−	−	+	−9.83	insig	−9.83	insig	3.49	4.68		−1.18
Cmdp005440	9.31E + 02	4.27E + 02	9.30E + 02	kingianoside G	+	−	−	+	−	−9.84		insig	9.06	−9.84		−9.06	−9.84
Cmdn006559	9.29E + 02	7.67E + 02	9.30E + 02	kingianoside I	+	−	−	+	+	−9.68		insig	11.85	−1.84	7.84	−11.85	−9.68
Zmcp005839	1.02E + 03	4.15E + 02	1.02E + 03	Parisyunnanoside B	+	+	−	+	+	−16.57	−16.35	3.13	3.35	insig	insig	−19.69	insig
Zmcp005801	1.03E + 03	4.15E + 02	1.03E + 03	Diosgenin-3-*O*-glcosyl(1→4)rhamnosyl(1→4)rhamnosyl(1→2)glcoside	+	+	−	+	+	−15.90	insig	insig	10.87	−8.08	insig	−17.87	−8.9
Cmlp005341	1.03E + 03	8.69E + 02	1.03E + 03	Pseudoprotodioscin	+	−	−	+	−	−16.65		insig	18.47	−16.65		−18.47	−16.65
Jmyp4441	1.05E + 03	4.15E + 02	1.05E + 03	(3β,14α)-3-*O*-β-d-glucopyranosyl-(1→2)-[β-d-xylopyranosyl-(1→3)]-β-d-glucopyranosyl-(1→4)-β-d-galactopyranoside-yamogenin	+	+	+	+	+	−6.62	−6.32	−2.25	−1.95	insig	insig	−4.37	insig
Hmqp005575	1.05E + 03	4.15E + 02	1.05E + 03	Aculeatiside A	+	+	+	+	+	−6.58	−6.59	−2.01	insig	insig	insig	−4.57	insig
Jmyp4236	1.06E + 03	4.15E + 02	1.06E + 03	polygonatumoside F	+	+	+	+	+	insig	insig	2.87	3.62	−1.06	insig	−3.23	insig
Jmyp3897	1.08E + 03	4.13E + 02	1.08E + 03	(25s)-spirost-5en-3β,14α-diol-3-*O*-β-d-glucopyranosyl-(1→2)-[β-d-glucopyranosyl-(1→3)]-β-d-glucopyranosyl-(1→4)-β-d-galactopyranoside	+	+	−	+	+	−11.51	−12.14	insig	insig	5.28	4.65	−10.03	insig
Jmyp3798	1.21E + 03	5.91E + 02	1.21E + 03	Polygonatumoside A	+	+	+	+	+	3.92	5.37	insig	2.05	insig	2.43	3.33	−1.45

+, detected; −, not detected; log_2_(fold-change) in columns 11–18, positive and negative values respectively represent increases and decreases, whereas insig indicates a lack of a significant change.

**Table 3 molecules-27-01445-t003:** OPLS-DA parameters for predicting differentially abundant metabolites.

Group Name	R2X	R2Y	Q2
Ps1_vs._Pc	0.806	1	0.966
Ps1_vs._Pk	0.834	1	0.990
Ps1_vs._Pm	0.717	1	0.965
Ps2_vs._Pc	0.829	1	0.989
Ps2_vs._Pk	0.790	0.999	0.984
Ps2_vs._Pm	0.759	1	0.941
Ps1_vs._Ps2	0.650	1	0.868
Pk_vs._Pc	0.809	1	0.966

**Table 4 molecules-27-01445-t004:** Comparative analysis of differentially abundant metabolites among five seed samples.

Group Name	Number of Insig Diff	Number of Sig Diff	Down	Up	Sig Diff %
Ps1_vs._Pc	329	522	90	432	61.34%
Ps1_vs._Pk	377	483	99	384	56.16%
Ps1_vs._Pm	486	344	122	222	41.45%
Ps2_vs._Pc	365	480	98	382	56.80%
Ps2_vs._Pk	410	440	101	339	51.76%
Ps2_vs._Pm	511	315	143	172	38.27%
Ps1_vs._Ps2	591	221	65	156	27.22%
Pk_vs._Pc	462	401	179	222	46.47%

## Data Availability

All data are provided in the manuscript.

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
