# Peer review of "Untargeted Metabolomics Analysis Revealed the Major Metabolites in the Seeds of four Polygonatum Species"

_molecules, 2022, doi:10.3390/molecules27041445_

Round 1

Reviewer 1 Report

This is a review for the manuscript “Untargeted metabolomics analysis revealed the major metabo-2 lites in the seeds of four Polygonatum species.” The aim of the study is to analyze the metabolic profiles of the seeds  of four Polygonatum species and to verify their medicinal and nutritional value via comparative analyses.

Lines 19-20,

“ The results of principal component analysis and hierarchical clustering were consistent indicating that the metabolites in Ps and Pm are similar, but differ greatly from Pc and Pk. This result is also consistent with the geographical distribution of the four Polygonatum plant species.” I found this statement is not insightful and confusing. The geographic correlation and the chemical components correlation is meaningless. What the authors imply from this observation?

Lines 74-76,

“For our metabolomics analysis, we used a UPLC-QTOF-MS/MS system to explore the seed chemical compositions and to verify the medicinal value or edibility of the examined species.” How examine seeds can lead to evaluate edibility of the examined species as a whole? Examine seeds can only lead to the conclusion about the seeds.

Lines 84-85,

It is not directly clear that the 873 metabolites are unique ones or redundancy is counted.

Discussion:

The discussion is general not specific about the results of this study and comparison of previous work on these species studied here.

Furthermore, lines 274-280, it said that the phylogenetic relationships were consistent with the results of the metabolomics analysis. But there is no evidence discussed here show that consistency. Please clarify and cite/discussion the particular findings from the cited article to show the consistency. Also, in the abstract, the authors should correlated the results of the metabolomics analysis to the phylogenetic relationships of the four species instead of the geographic distribution.

  1. Floden, A.; Schilling, EE. Using phylogenomics to reconstruct phylogenetic relationships within tribe Polygonateae (Asparagaceae), with a 420 special focus on Polygonatum. Molecular Phylogenetics and Evolution.2018, 129, 202-213. 421
  2. Meng, Y.; Nie, Z.L.; Deng, T.; Wen, J. Phylogenetics and evolution of phyllotaxy in the Solomon’s seal genus Polygonatum (Asparagaceae: 422 Polygonateae).Bot. J. Linn. Soc. 2014,176, 435-451. 423

Reviewer 2 Report

The work entitled Untargeted metabolomics analysis revealed
the major metabolites in the seeds of four Polygonatum species,
submitted by J. Qi and collaborators, describes the composition of
the seeds of 4 species of this genus, using a metabolomic analysis
based on UPLC-QTOF-MS/ MS. The work contains a competent and well-executed experimental part,
the only recommendation of this reviewer is to shorten the introduction
of the manuscript, since most of the information presented is about
the studies carried out on the rhizomes of these plants, which are
widely used in traditional Chinese medicine. It is recommended to reduce
the introduction and emphasize that the study of seeds is very limited
and that this work contributes to the knowledge of the chemical
composition of this part of the species of the genus Polygonatum.
